

# Lagged variation of moisture conditions in central Asia compared with monsoonal Asia during the last four interglacials

Jia Jia[1], Jianhui Chen[2,*], Xin Wang[2], Hao Lu[2], Zhiyuan Wang[1], Zaijun Li[2], Qiang Wang[2], Yanwu Duan[2], Ilhomjon Oimahmadov[3], Mustafo Gadoev[3], Fahu Chen[4,2]

[1]College of Geography and Environmental Science, Zhejiang Normal University, Zhejiang 321004, China

[2]MOE Key Laboratory of West China's Environmental System, College of Earth and Environmental Sciences, Lanzhou University, Lanzhou 730000, China

[3]Institute of Geology, Tajik Academy of Science

[4]Chinese Academy of Sciences Center for Excellence in Tibetan Plateau Earth Sciences, Beijing 100101, China

* Corresponding Authors: Jianhui Chen (Email: jhchen@lzu.edu.cn)

**Abstract**: Previous research has indicated that variations in moisture conditions in arid central Asia (ACA) were out-of-phase with those of monsoonal Asia during the Holocene. In order to investigate this phenomenon, we compared the pattern of moisture variations in ACA and the region dominated by the East Asia summer monsoon (EASM) during the last four interglacials. The results indicate that moisture variations (pre) in ACA lagged those in the EASM region by 3 kyr during MIS 5, by 0 kyr during MIS 7, by 2 kyr during MIS 9, and by 5 kyr during MIS 11. We suggest that this lagged pattern in three out of four interglacials was the result of a zonal climatic teleconnection, westerly wind intensity, and evaporation upstream. Overall, our results shed new light on the climatic variability of central Asia and its origins during the Holocene.

## 1. Introduction

Asia can be climatically divided into two regions: monsoon-dominated Asia which is



characterized by a humid environment, and arid central Asia (ACA) which is
characterized by an arid environment. The climate of ACA, a part of
westerlies-dominated Asia, including the greater part of central Asia, northeastern Iran,
and Xinjiang province in China, has shown an opposite pattern of variation to that of
Southern Europe and North-central China over the last few decades (Huang et al.,
2015). On the millennial scale, geological records also indicate that moisture
variations also exhibited an anti-phased pattern of variation between arid central Asia
(ACA) and monsoon-dominated Asia, which is typified by a wet "Little Ice Age" and
a dry "Medieval Climatic Anomaly" (Chen et al., 2010a). In addition, on the
multi-millennial scale, loess records reveal a persistent wetting trend during the
Holocene - following a wet early Holocene - in the regions dominated by the Indian
summer monsoon (ISM), and a wet mid-Holocene in the regions dominated by the
East Asian summer monsoon (EASM) (Wang et al., 2013; Chen et al., 2016).
Moreover, several studies have indicated that similar phenomenon occurred during
previous interglacials (e.g. Huang et al., 2015; Chen et al., 2016).

In order to examine the consistency of this anti-phased pattern of behaviour between
ACA and the adjacent regions, we previously investigated the last interglacial, with a
duration of more than 50 kyr and which included two and a half precession cycles.
Our results indicated that moisture variations in ACA lagged those of the
EASM-dominated regions by 3-5 kyr (Jia et al., 2018a). In the present study, we
extend the analysis of this relationship to the last four interglacial periods.

During the Quaternary two major climatic transitions occurred which have attracted
major research attention: the mid-Pleistocene transition (MPT) at 0.8 Ma and the
mid-Brunhes event (MBE) at 0.43 Ma. The MPT was characterized by an important
shift in global climate evolution from the previous dominant 41-kyr climatic cyclicity
to the subsequent dominant 100-kyr cyclicity (Ruddiman et al., 1986; Berger, 1989;
Shackleton et al., 1990). The MBE was characterized by the complete establishment
of high amplitude 100-kyr climatic cyclicity with much warmer interglacials and
cooler glacials then previously (e.g. EPICA community members, 2004; Lisiecki and
Raymo, 2005). The four interglacials which are the concern of the present study thus





had similar boundary conditions to the modern Holocene interglacial and for this
reason they were selected for detailed investigation.

2. Geological setting and the studied section

ACA is far from oceanic moisture sources and is therefore an arid environment. The

western part of ACA contains widespread sandy desert, while the eastern part is

characterized by a basin-mountain topography (Fig. 1). Precipitation occurs mainly in

the mountains and adjacent areas and rarely in the basins. In the Junggar Basin, the

mean annual precipitation (MAP) is less than 50 mm, but gradually increases to more

than 1000 mm on the northern slopes of the North Tienshan Mountains. The rivers

rise in the high mountains and flow into the lakes in the arid basins.

Figure 1

Today ACA is climatically dominated by the westerlies with precipitation in most

regions predominantly in winter-spring; only in the northern part is the precipitation

predominantly in summer-autumn. Notably, carbon isotope records indicate that a

continental dry summer climate was established by at least 1.77 Ma (Yang and Ding,

2006).

Loess deposits are one of the most important geological archives in the region (e.g.

Ding et al., 2002; Yang et al., 2006; Chen et al., 2016; Jia et al., 2018b). They are

widespread on alluvial fans, river terraces, and on the piedmont slopes of the Tienshan

and Pamir mountains (Frechen and Dodonov, 1998; Sun et al., 2002; Li et al., 2018),

and they have preserved paleoclimatic records from the early- to the late Pleistocene

(Frechen and Dodonov, 1998; Ding et al., 2002; Wang et al., 2018; Li et al., 2019).

The most complete published loess record was obtained from Tajikistan (Frechen and

Dodonov, 1998; Ding et al., 2002). In the present study, the Darai Kalon (DK) section

was selected to retrieve a record of moisture variations of the last four interglacials.

The Holocene (modern interglacial) was excluded from the study, since this interval

may be eroded or partly eroded, as suggested by Frechen and Dodonov (1998).





The DK section (38º23′4″N, 69º50′1″E, 1561 a.m.s.l.; Fig. 1) is 176 m in thickness
and contains 29 paleosols, according to the investigation of Ding et al. (2002). The
upper four paleosols were selected in the present study. Paleosols S1, S2, and S3 are
pedocomplexes, which comprise 3, 2 and 2 soil layers, respectively. S4 consists of a
single soil layer. The soil layers are separated from the underlying less-weathered
parent material by a thin carbonate horizon. A detail stratigraphic description is given
elsewhere (Dodonov et al., 2006; Jia et al., 2018b).

It is widely observed that the loess of the Chinese Loess Plateau (CLP) provides a
continuous and long-term record of fluctuations in moisture conditions in the
EASM-dominated region during the Quaternary (e.g. Ding et al., 1995; Guo et al.,
2009; Lu et al., 2018). The Xifeng (XF) section (35º45′31″N, 107º41′45″E, 1345
a.m.s.l.; Fig. 1), located on the northwest edge of the EASM-dominated region, is
acknowledged as preserving the most complete Quaternary record (e.g. Guo et al.,
2009; Hao et al., 2012; Lu et al., 2018). In this study, the upper four loess-paleosol
alternations were investigated for comparison with the loess record from Tajikistan.

## 3. Chronological framework

Grain size analysis was conducted on all of the samples using the methods of Lu and
An (1997). After sequential removal of organic matter with 10% $H_2O_2$ and carbonate
with 10% HCl, and dispersal using 0.05 N $(NaPO_3)_6$, the samples were measured
using a Mastersizer 2000 laser diffraction particle size analyzer with size range of
0.02-2000 μm.

The eolian mineral dust comprising the loess of the CLP is transported by the East
Asian winter monsoon (Guo et al., 2009). Investigations of Chinese loess have
revealed a close link between the grain size of loess to variations in Northern
Hemisphere ice sheets effected via the Siberian High anticyclone (e.g. Ding et al.,
1995; Guo et al., 2009; Hao et al., 2012, 2015). Hao et al. (2012) confirmed that this
close coupling between high northern latitude cooling and increased dust activity in
the deserts of the Asian interior deserts operated on timescales ranging from decadal
to Earth orbital. Therefore, the grain size of Chinese loess provides independent





evidence for ice volume changes in the Northern Hemisphere. Importantly, the loess

in ACA has similar paleoenvironmental implications as Chinese loess (Ding et al.,

2002). The grain-size variations of the loess of ACA and the CLP loess correspond

closely to the deep sea benthic $\delta^{18}O$ curve via their common linkage with Northern

Hemisphere ice volume. Therefore, the chronology of the DK and XF loess sections

can be established using the accepted correlation scheme between the loess grain-size

record of loess and the benthic $\delta^{18}O$ record of marine sediments.

The age control points are shown in Figure 2, which are locating the boundary of

Marine Isotope Stage (MIS). By determine the sample location with mean value

between peak and valley around soil/loess boundary, we obtained the age control

points in DK and XF section. The Linear interpolation between age control points was

then used to generate a final timescale. The depths and age control points are listed in

Table 1. In addition, due to high climate resolution recording by Last Glacial (LG)

loess in DK section, its chronology was construct by matching the NGRIP curve. The

result had been published in Wang et al. (2018). Based on the resulting chronologies,

grain-size time series are presented in the Figure 3, which demonstrate that the

variations in grain-size in ACA and the CLP are synchronous, which is supported by

the results of cross correlation analysis (Fig. 4).

Table 1

Figure 2

Figure 3

Figure 4

4. Climatic proxies and their implications

After air-drying in the laboratory, 5.5 g of powder sediment was packed into 10 ml

plastic boxes and used for magnetic susceptibility measurements. Magnetic



susceptibility was measured at 470 Hz and 4700 Hz ($\chi_{lf}$ and $\chi_{hf}$, respectively) using a Bartington Instruments MS2B sensor. Frequency-dependent magnetic susceptibility ($\chi_{fd}$) was calculated as $\chi_{fd} = \chi_{lf} - \chi_{hf}$.

The four major magnetic minerals in loess are hematite, goethite, magnetite, and maghemite (e.g. Maher, 1998; Liu et al., 2007). It had been widely observed that weakly magnetic hematite and goethite only make a small contribution to the magnetic susceptibility, while in contrast strongly magnetite and maghemite, although present in trace contents, make a large contribution (e.g. Liu et al., 2007; Wang et al., 2018). $\chi_{fd}$ is extremely sensitive to the fine-grained ferrimagnetic component of loess (Liu et al., 2007), which is 20-30 nm maghemite (Liu et al., 2007; Chen et al., 2010b). According to magnetic and mineralogical evidence, it has been proposed that the fine-grained maghemite is pedogenic, and its content can be used as proxy of paleo-precipitation (e.g. Maher et al., 1995; Jia et al., 2013; Song et al., 2014). Therefore, the high $\chi_{fd}$ of soil units indicates favorable soil forming conditions under a prevailing humid climate, and the low $\chi_{fd}$ of soil units indicates soil formation under a prevailing dry climate.

## 5. Results

The $\chi_{fd}$ of the DK section varies within the range of $0.0\text{-}18.5\times10^{-8}\text{m}^3\text{kg}^{-1}$ and of the XF section it varies within the range of $0.8\text{-}26.2\times10^{-8}\text{m}^3\text{kg}^{-1}$ (Fig. 5a). Three peaks are evident in both $\chi_{fd}$ records during MIS 5, and one poorly defined peak is evident during 130-120 ka in the XF section; however, at least three soil units can be readily distinguished in the field. Since loess is also a typical eolian dust, pedogenesis must be influenced by the processes of dust deposition (such as dust accumulate rate) as well as by the local climate. In order to minimize the influence of dust deposition and make the pedogenic signal more obviously, the FFT filtering analysis has been applied on the $\chi_{fd}$ curves. Due to the development of soil units contains strongly precession component, such as: three soil units developed during MIS 5, at least two soil units developed during MIS 7, two soil units developed during MIS 9, we decided to filter out the precession cycle (19-23 kyr) to do the comparative analysis between DK and





XF records. The resulting curve exhibits three peaks which have an out-of-phase
pattern of variation compared with the DK loess record (Fig. 6a). Cross-correlation
analysis of loess records reveals that during 130-75 ka the precipitation variations in
ACA lagged that of the EASM-dominated region by 3 kyr (Fig. 6e). Similarly, from
the KS loess record, it can be seen that: the moisture variations lagged those of the
EASM-dominated region by ~3-5 kyr during MIS 5 (Jia et al., 2018a).

Both loess records exhibit two peaks during MIS 7 (Fig. 5b). In the DK section $\chi_{fd}$
varies within the range of 0.4-10.5×10$^{-8}$m$^3$kg$^{-1}$, and in the XF section it varies within
the range of 0.8-26.2×10$^{-8}$m$^3$kg$^{-1}$. The filtering curve exhibited that, during this period,
precipitation change in the DK section shows synchronous variation with that in XF
section (Fig. 6b), which supported by cross-correlation analysis (Fig. 6f).

Unlike MIS 5 and MIS 7, which span one and a half obliquity cycles or three
precession cycles, the duration of MIS 9 and 11 are much shorter, and they only
include one obliquity cycle or two precession cycles. During MIS 9, the $\chi_{fd}$ in the DK
section varies within the range of 1.4-18.5×10$^{-8}$m$^3$kg$^{-1}$, and within the XF section
within the range of 8.7-25.9×10$^{-8}$m$^3$kg$^{-1}$ (Fig. 5c). According to cross-correlation
analysis, the precipitation variations in the DK section during 290-345 ka lag those of
the XF record by 2 kyr (Fig. 6g). During MIS 11, the $\chi_{fd}$ in the DK section varies
within the range of 0.7-9.2×10$^{-8}$m$^3$kg$^{-1}$, and in the XF section within the range of
9.0-21.6×10$^{-8}$m$^3$kg$^{-1}$ (Fig. 5d). The variations of $\chi_{fd}$ curves are dominated by obliquity
(Fig. 5d). After application of 19-23 kyr band-pass FFT filtering, two peaks are
evident in both curves during (during the period 424-379 ka) (Fig. 6d).
Cross-correlation analysis of the loess records suggests that during 430-385 ka in
ACA, precipitation variations lagged those of the EASM-dominated region by 5 kyr
(Fig. 6g).

Figure 5

Figure 6





6. Discussion

Previous investigations of the Holocene in ACA have proposed that climate change in central Asia tracked insolation variations (an external factor) on the orbital scale (e.g. Ding et al., 2002; Bronger, 2003); however, climate change was also forced by a series of internal factors. Climate simulations indicate that the westerly wind intensity and upstream evaporation are the dominant factor determining humidity variations in

ACA (Jin et al., 2012). Huang et al. (2015) emphasized the effect of a zonal climatic teleconnection in which humidity variations were anti-phased between ACA and the Indian monsoon region. There are potentially two mechanisms which can generate such as anti-phased relationship: (1) A stronger (weaker) Indian summer monsoon (ISM) can lead to northward (southward) movement of the westerlies. (2) A stronger

(weaker) ISM can lead to an extended duration of the summer monsoon and a shorter duration of the westerlies influence in Tajikistan. Accordingly, a stronger (weaker) ISM results in a precipitation increase (decrease) in the ISM-dominated region, but the northward (southward) movement of the westerlies may result in a precipitation decrease (increase) in ACA.

Both the EASM and ISM are components of the Asian summer monsoon (ASM) system. However, precipitation varies in out-of-phase pattern between the EASM-dominated and ISM-dominated regions (e.g. Chen et al., 2016). Evidence from various geological archives provides strong support for this model. For example, records from the CLP exhibit synchronous but anti-phased relationships between the

summer and winter monsoon in East Asia, on timescales from multi-millennial to Earth orbital (e.g. Ding et al., 1995; Guo et al., 2009; Kang et al., 2018). In contrast, stalagmite records from South China, which have a robust chronology, document that the glacial termination of the Asian summer monsoon occurred 3 kyr earlier than that is evident in the global ice volume record (Cheng et al., 2009). In a review of previous

research, Wang and Liu (2016) proposed that stalagmite oxygen isotope record was dominated by variations in the ratio of moisture from the Indian Ocean and from the Pacific Ocean, rather than by precipitation. In a comparison with the Holocene moisture pattern in the East Asian monsoon margin, it was suggested that a major





proportion of the variance of the oxygen isotope record from Chinese caves was
contributed by the Indian Monsoon signal (e.g. Chen et al., 2015). Due to the close
link between the grain-size records from the CLP and Northern Hemisphere ice
volume variations, the previous results reveal an out-of-phase pattern of variation in
moisture conditions between the EASM-dominated and ISM-dominated regions. A
well-dated high-resolution lake sediment record from North China demonstrates that
the EASM-dominated region experienced a dry climate during the early Holocene,
whereas the ISM-dominated region experienced a humid climate (Chen et al., 2015).
The lagged response of Northern Hemisphere ice volume to insolation variations may
be the major cause of the lagged response of climate change in high and mid-latitudes
to insolation (e.g. Ding et al., 1995).

Combining the foregoing modelling and geological evidence, an out-of-phase
variation of moisture conditions between ACA and the EASM-dominated region is
indicated during interglacials. This scenario is supported by the results of the present
study, which indicate that during the last three out of four interglacials there was a lag
in moisture changes in ACA compared to the EASM-dominated region. In addition,
our results contribute to an improved understanding of climate change in ACA.
During MIS 9, which was a relatively cool interglacial during which the ASM was
relatively weak (e.g. Guo et al., 2009; Hao et al., 2012), there was a relatively brief 2
kyr delay in moisture change. In contrast, during MIS 11, a warm interglacial with a
relatively strong ASM (e.g. Guo et al., 2009; Hao et al., 2012), there was a much
longer delay of 5 kyr. Our results indicate that the length of the lag was variable and
related to the intensity of the ASM, with a stronger ASM corresponding to longer lag
and a weaker ASM corresponding to a short lag. This phenomenon is well explained
by the foregoing model.

Among the studied interglacials, MIS 7 is distinguished by synchronous moisture
variations on precession component. As illustrated in Figure 7, an interglacial climate
is normally characterized by a rapid increase of precipitation at the beginning with a
subsequent gradual decrease (Fig. 7a-d). However, the records from the CLP indicate
the reverse pattern of climatic variation: a gradual increase from the beginning of the





interglacial and a rapid decrease at the end (Fig. 7d). Furthermore, the MIS 7 is the

coolest interglacial among past four interglacials. According to the conceptual model,

a gradually strengthening ASM and the relatively weak ASM in MIS 7 are both

beneficial to westerlies staying in ACA during the early stage, rather than northward

movement. For this reason, the moisture record in ACA exhibits a synchronous

humidity variation compared to the EASM-dominated region.

Figure 6

7.  Conclusion

We have investigated the timing of changes in moisture conditions in ACA during the

last four interglacials. The results show that changes in moisture conditions in ACA

lagged those of the EASM-dominated region by 3 kyr during MIS 5, by 0 kyr during

MIS 7, by 2 kyr during MIS 9, and by 5 kyr during MIS 11. These findings support

our previous conclusions regarding the timing of climate change in ACA during the

Holocene and MIS 5 (e.g. Chen et al., 2016, 2019; Jia et al., 2018a), and they also

show that length of the lag was variable and influenced by the intensity of the ASM

and especially the ISM. The results also support the concept of a zonal climatic

teleconnection (Huang et al., 2015), which results in the westerly wind intensity and

evaporation upstream (Jin et al., 2012) dominating changes in humidity conditions in

ACA during interglacials.

Acknowledgements: This work was supported by the National Key Research &

Development Program of China (2018YFA0606401) and National Natural Science

Foundation of China (grants 41771213, 41822102).

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





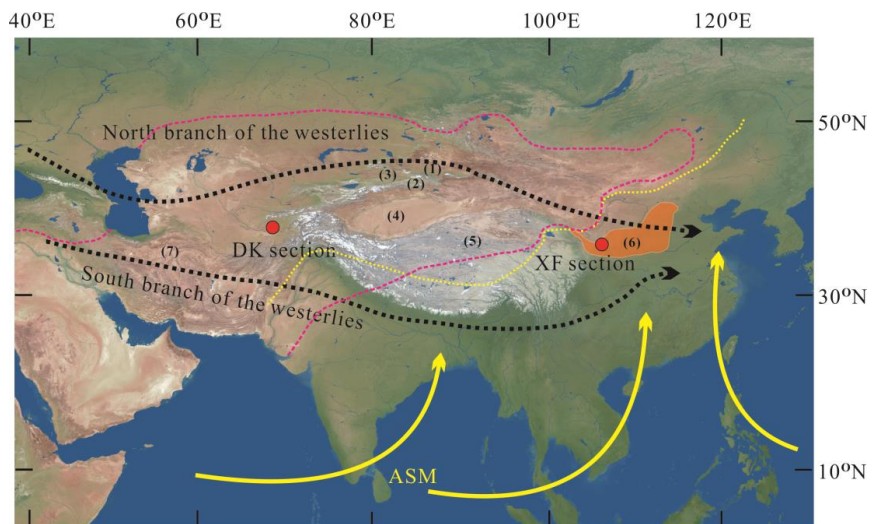

Figure 1. RS image of arid central Asia (ACA) and Monsoon Asia with the pink dashed line indicating the present Asian monsoon limit. The area enclosed by the yellow dashed line is ACA (modified from Huang et al., 2015). The number 1-6 are Junggar Basin, Tienshan Mountains, Ili River Basin, Tarim Basin, Tibet Plateau, and Chinese Loess Plateau, as sequences. The range of Chinese Loess Plateau is indicated by orange shadow.

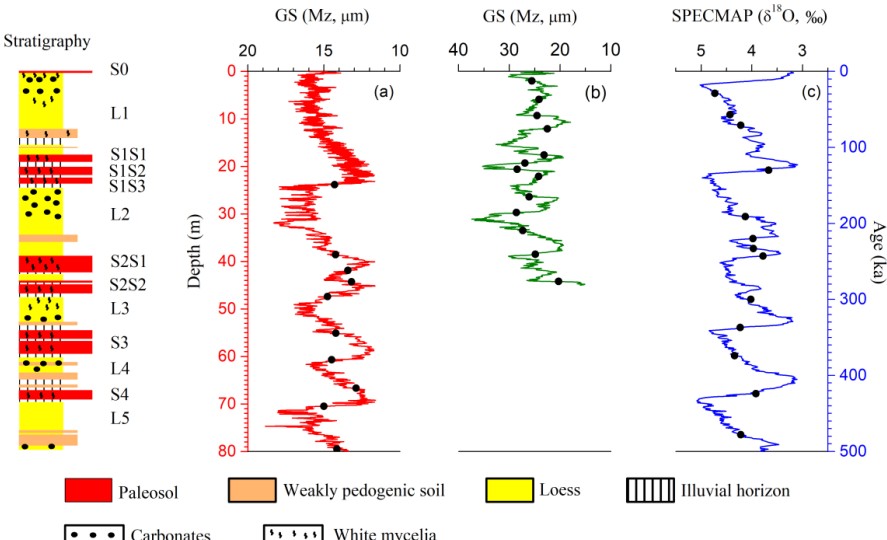

Figure 2. Pedostratigraphy of the DK section (Jia et al., 2018a). The comparisons of
climate variations among grain size record in DK section (a), grain size record in XF
section (b, Lu et al., 2018), and SPECMAP $\delta^{18}$O record (c, Lisiecki and Raymo, 2005).
The black dots in the curves are the location of age control points.

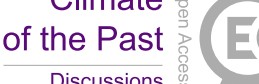

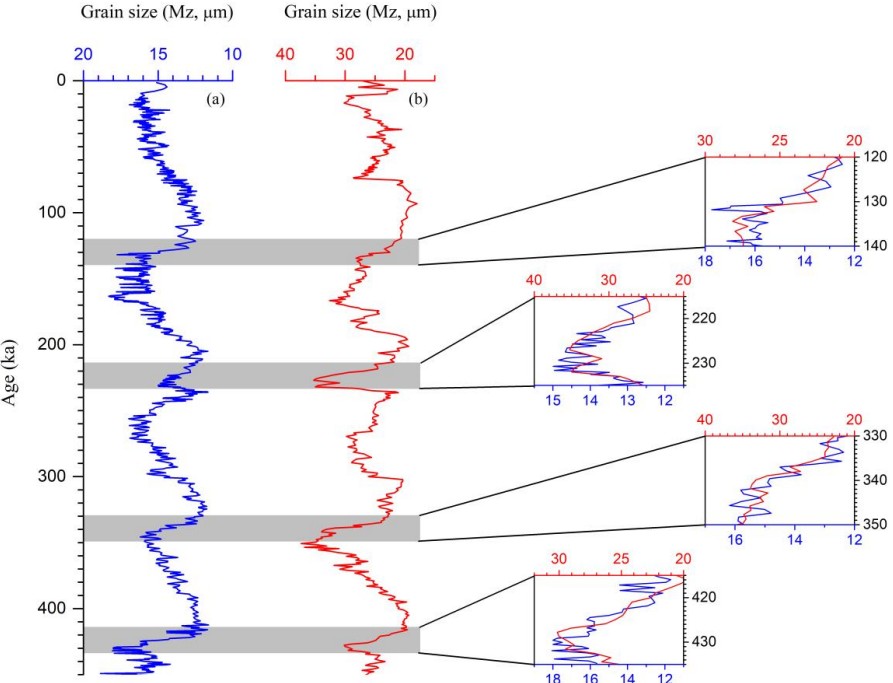

Figure 3. Comparison of variations in mean grain-size in the DK section (a, blue curve) and the XF section (b, red curve). The enlarged sub-plots show detailed comparisons of the initiation of four interglacials (MIS 5, 7-1, 9, and 11).





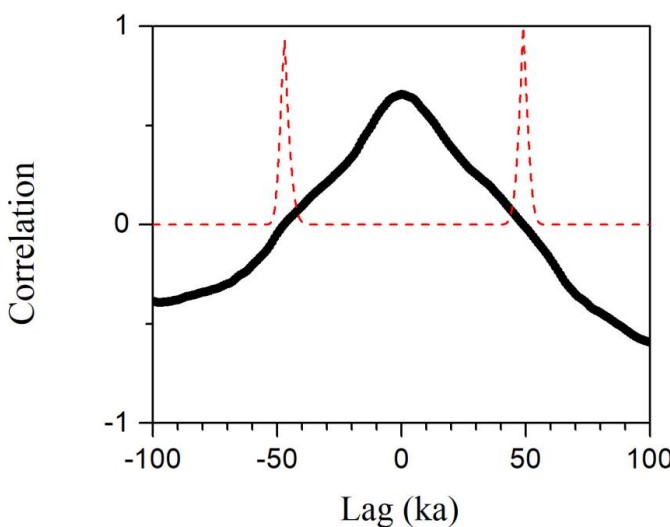

Figure 4. Results of cross-correlation analysis of the grain-size records from the DK and XF

sections. The red curve is the r value.



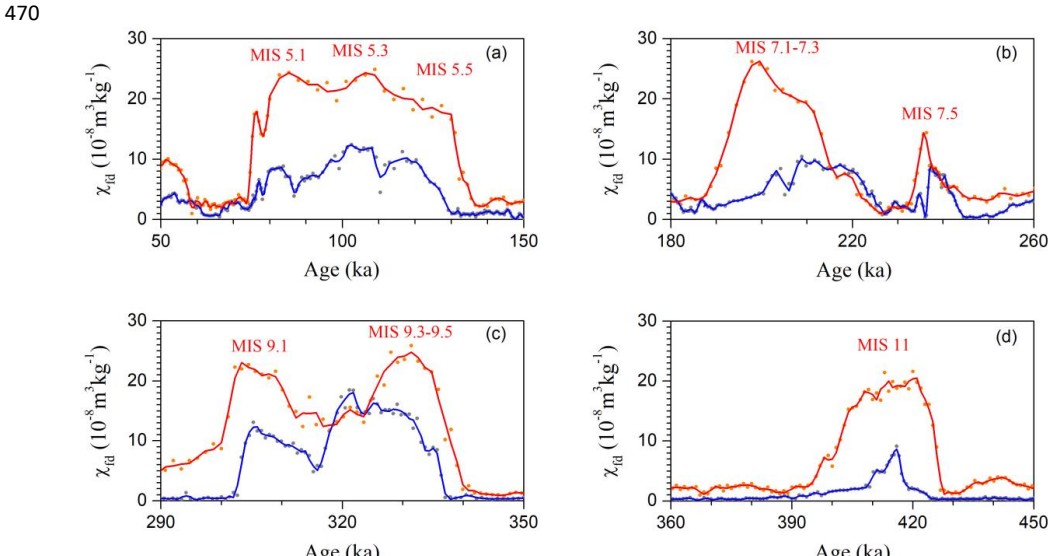

Figure 5. Comparison of records of frequency-dependent magnetic susceptibility ($\chi_{fd}$) for four interglacials in the DK section (blue curve) and XF section (red curve). The dots are the measured data, and the lines are the results of five-point smoothing. (a) Comparison during

MIS 5, (b) comparison during MIS 7, (c) comparison during MIS 9, (d) comparison during MIS 11. The data for XF are from Lu et al. (2018).

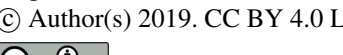



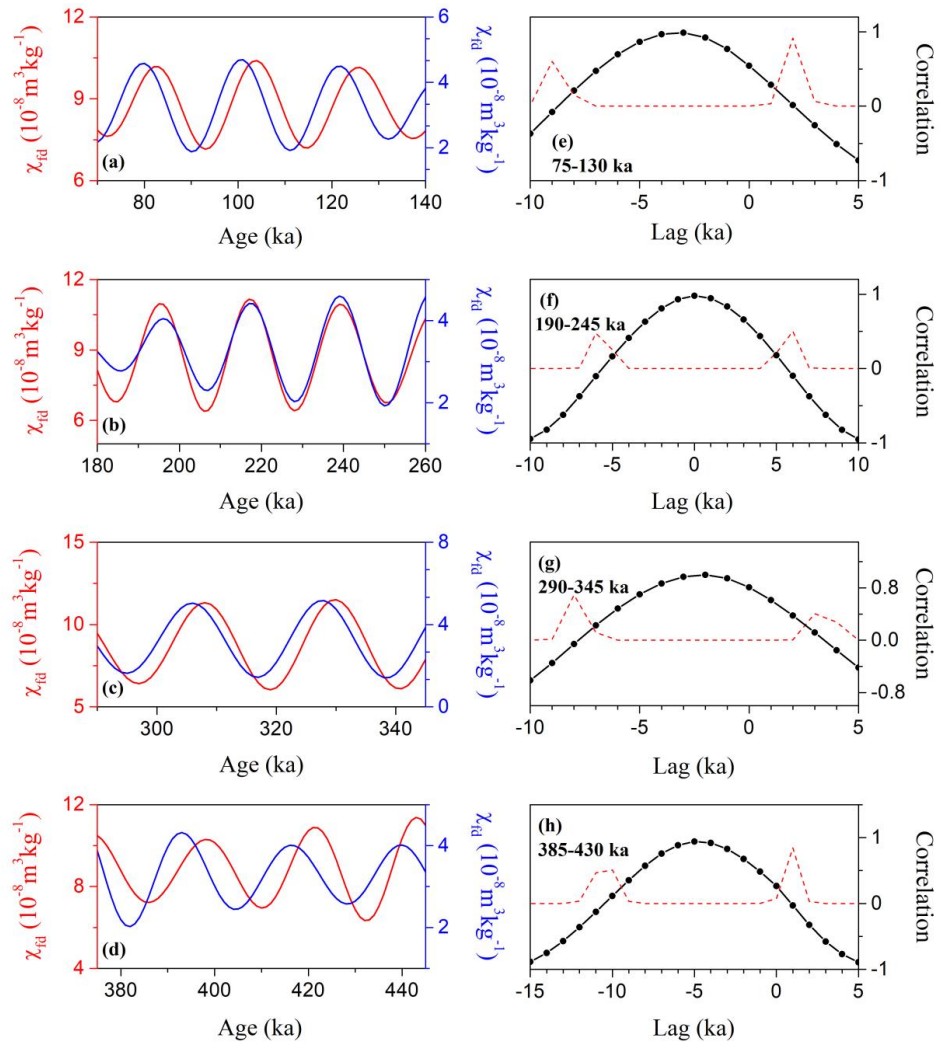

Figure 6. Comparison of records of 23 kyr high-pass FFT filtered frequency-dependent magnetic susceptibility ($\chi_{fd}$) during four interglacials. (e-h) Cross-correlation analysis of precipitation components from the DK and XF records. The red curves in figures (e-h) are r values





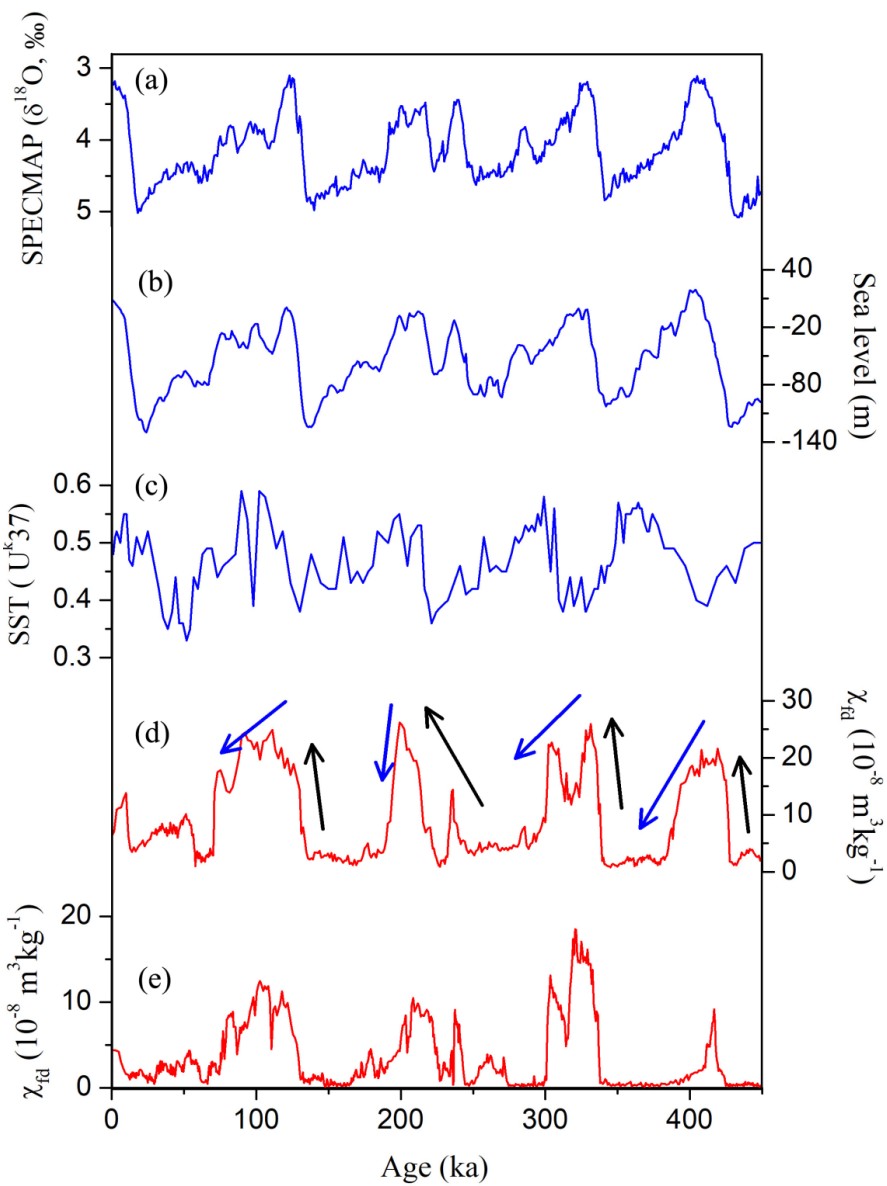

Figure 7. Comparison of patterns of interglacial climatic variability recorded by the marine oxygen isotope record of global ice volume (a, Lisiecki and Raymo, 2005), global average sea level (b, Spratt and Lisiecki, 2016), North Atlantic sea surface temperature (c, Lawrence et al., 2009), and the frequency-dependent magnetic susceptibility record ($\chi_{fd}$) of the Chinese Loess Plateau (d, Guo et al., 2009), and the Tajikistan loess (e).





Table 1. Age control points and corresponding depths for the DK loess section.

| Age (ka) | Depth (m) | | Age (ka) | Depth (m) | |
|---|---|---|---|---|---|
| | DK section | XF section | | DK section | XF section |
| 0 | | 0 | 233 | 44.3 | 20.6 |
| 11.5 | | 0.7 | 243 | 47.3 | 22.1 |
| 29 | | 2 | 300 | 55.1 | 26.4 |
| 57 | | 5.9 | 337 | 60.7 | 29.7 |
| 74 | | 9.2 | 374 | 64.7 | 33.5 |
| 130 | 23.85 | 12.1 | 424 | 70.5 | 38.5 |
| 191 | 38.6 | 17.6 | 478 | 79.5 | 44.2 |
| 220 | 41.9 | 19.3 | | | |

495