# Peer review of "Lagged variation of moisture conditions in central Asia compared with monsoonal Asia during the last four interglacials"

_Climate of the Past, 2019_

## Short Comment (SC1) · 3 Dec 2019

Using high quality, high resolution grain size/magnetic susceptibility records of the Darai Kalon/Chasmanigar (DK, Tajikistan) and Xifeng (XF, China) loess sequences Jia et al. suggest that moisture variations in Arid Central Asia (ACA) lagged those in monsoonal Asia by 2-5 ka over the last 4 interglacials. These lags were quantified using chronologies that were developed via tuning the loess records to the SPECMAP (and partly NorthGRIP-Greenland) curves, i.e. the authors do not provide independent absolute chronologies for the investigated records. This implies that they effectively dis-regard potential regional differences in climate response during the tuning procedure

(this is a fundamental assumption of tuning), and yet they then use the tuned records to quantify lags between the records, i.e. the difference in moisture response of these two regions. In my opinion, while the authors provide great proxy records from the DK/XF sections, the methodology is circular and flawed. Leads-lags between records can only be properly estimated using high-resolution, absolute independent loess chronologies such as those presented in Stevens et al. (2018) and very few other papers. The authors should also consider the uncertainties inherent in the tuning process, which are not assessed at all in this manuscript. Clearly, the SPECMAP chronology has a non-negligible uncertainty, and the tuning procedure as well, and the resultant combined error will no doubt exceed many of the 'lags' reported here. Furthermore, taking the mean values of maxima and minima to identify tie-points does not seem to be the best choice in this case, or at least this approach has not been justified. Usually the first derivatives of the proxy records are taken as tie-points, as this effectively reveals the inflection points of the "proxy function", where the proxy time series changes the most.

In short, the method of age model construction does not permit the interpretation presented here. Considering these purely methodological concerns, I would not recommend this manuscript for publication in CP in its present form.

References

Stevens T, Buylaert J-P, Thiel C, Újvári G, Yi S, Murray AS, Frechen M, Lu H, 2018. Ice-volume-forced erosion of the Chinese Loess Plateau global Quaternary stratotype site. Nature Communications 9, 983.

---

## Referee Comment (RC1) · Anonymous Referee #1 · 6 Dec 2019

The study by Jia et al. deals with an interesting loess-palaeosoil section in Central Asia (Darai Kalon) with the concern to reconstruct palaeoenvironmental conditions for the last four glacial-interglacial cycles. Furthermore, the results are compared to a loess section from the Chinese Loess Plateau (Xifeng) in order to find spatial and temporal relations between humidity periods and atmospheric circulation patterns, especially related to the interaction between westerlies, the Indian Summer Monsoon and the East Asian Summer Monsoon.

In its present form, this study reveals some weak points that mainly relate to the methodological approach for establishing chronologies for the loess sections.

[Figure]

(I) First, no independent age determination is presented that could support the chronology that is solely based on the assumed relation between grain sizes and d18O values from marine cores. Thus, the whole chronology remains very speculative. If the chronologies of loess sections from different areas are both established based on a peak matching between the grain-size curves and one d18O-curve, of course the resulting curves are synchronous - that is circular reasoning.

(II) Second, the direct correlation between grain-size values / magnetic parameters and the Specmap-isotope curve implies that the built-up of the curves (in the sense of the built-up of the loess-palaeosoil sequences from bottom to top) is a continuous, successive and more or less linear process. But loess sedimentation rates are highly variable and may even tend towards zero during interglacial periods (see e.g. the study by Frechen and Dodonov (1998) at the same section, cited in the manuscript). That means that phases of lacking (or strongly reduced) loess deposition that relate to periods of pedogenesis (that may extend over several thousands of years) are ignored in the age model in favor of producing a kind of continuous "loess graph" assuming uniform deposition rates between the time control points. In my opinion, beside all other uncertainties of the chronology, already this linear interpolation between a few age control points is a strong contradiction and produces large errors. This is all the more serious if the interpretation strongly relies on assumed time lags in the range of a few thousand years. Furthermore, pedogenic clay formation that leads to finer grain sizes in subsoil horizons is a secondary process that is decoupled from primary aeolian deposition, but it strongly affects the mean grain-size curve. The approach to directly correlate the gradual fining of the grain-size in a loess depth profile with a linear progression of time (e.g. sedimentary d18O-curves, SPECMAP) is far away from the general concept of the way in which soils are formed. In my opinion, this also applies to magnetic susceptibility curves as soon as it is assumed that increasing values are caused by pedogenic processes instead of sedimentary processes. Such an approach could be just about acceptable for the aim to e.g. make a general charcterisation of soil formation or for realizing a rough comparison or correlation of different soils from

different sections, but not for generating a precise chronology for pedogenically altered loess sections.

(III) Third, as I understand it, the interpretation of time lags is strongly based on the respective position and the possible offset between peaks of the grain-size and frequency dependant susceptibility curves. Or alternatively, if a palaeosoil shows a thicker subsoil horizon and a less rapid clay decline with depths, also the frequency-dependant susceptibility may show a less rapid decline as these values are generally strongly correlated with clay contents. This may lead to a different or shifted slope of the susceptibility curve. In that case the assumed time lags are again caused by pedogenic processes, i.e. different soil development depths. Therefore, despite value curves/ peaks of different sedimentological/pedogenetic parameters appear in different soil depths, both may relate to the same formation time. Without showing the proxy-curves in a clear stratigraphic context, it is not possible to evaluate the plausibility of the used approach. Furthermore, it would be necessary to show the strength of correlation between clay and frequency-dependant susceptibility (FDS) values, because if they show a strong correlation, FDS is not an independent value and thus even more severely influenced by the tuning process.

Finally, considering these mentioned points that all concern the reliability of the age model, which in turn is the base for the interpretation, I cannot recommend the manuscript for publication in "Climate of the Past" in its present form.

---

## Referee Comment (RC2) · Anonymous Referee #2 · 16 Dec 2019

In this manuscript, the authors compare the magnetic susceptibility records between the DK and the Xifeng sections and conclude that the moisture variation in central Asia lagged the one in monsoon Asia by 0-5 kyr during four interglacials. Understanding the phase relationship between climates of different regions is extremely important for improving our understanding of the forcing and feedback mechanisms, and this kind of study should be encouraged. However, the analysis of phase relationship between two records has a high demand on the quality and accuracy of the chronology of the records. In this study, there are weaknesses in chronology that prevent me to be convinced of the results.

Firstly and most critically, age uncertainty is not mentioned in the paper. As far as I know, the age uncertainty in loess records can reach several kyr. How would the conclusion of the authors be influenced by the age uncertainty of the two loess records?

Secondly, the chronology of both DK and Xifeng sections was developed by correlation with the marine oxygen isotope record. This is quite acceptable in many loess studies, but I feel it is questionable when discussing phase relationship because the chronology of the two records is not independent.

Lastly but less concerned, the mechanisms given by the authors to explain the phase difference between the two records is unclear for me.

In summary, the climate difference between Central Asia and monsoon Asia is an interesting topic and deserves to be better studied, but I don't suggest the authors to discuss about the phase relationship based on the data they have now.

——————————————

---

## Short Comment (SC2) · 13 Jan 2020

We agree with Dr. Ujvari's opinion that it is a better choice to date the chronology with a high-resolution, absolute independent dating method, if there has any select object for our research object. OSL is a good meter to determine the age of last interglacial aeolian sediment, and maybe penultimate interglacial aeolian sediment (Stevens et al., 2018). Our previous study had focused on the Holocene climate variation pattern comparison between Monsoon Asia and East Asia. The OSL data suggested the climate variation pattern of Monsoon Asia was lagging that of East Asia about 3-5 ka (Chen et al., 2016). However, in this study, we focus on the comparison of the last

four interglacials due to we believe the more objects we study, the closer we get to the truth. On this scale, the OSL dating result commonly presents a wide uncertainties range. Even for last interglacial and penultimate interglacial, the OSL uncertainty range can reach to 5 ka in a professional and excellent OSL study (Stevens et al., 2018), the length of which is similar to the lagging time referred in this paper. Due to the absolute independent dating method present a unignore uncertainty range, in this study, we selected a relatively dating method to further determine the chronology. The DK section had done the OSL and Paleomagnetic dating work in previous studies (Frechen and Dodonov, 1998; Ding et al., 2002). Base on those, we further developed the SPECMAP tuning chronology. We believe the absolute independent dating method combining with the relative dating method can produce a good chronology. Therefore, both two loess sequences were developed chronologies via tuning the particle size of loess to the SPECMAP (and partly NorthGRIP-Greenland) curves in this study. That is a commonly used chronology method for mid-Pleistocene loess (e.g. Sun et al., 2006; Guo et al., 2009; Hao et al., 2012). As you mentioned, this method may lead to a uncertainty range. However, this uncertainty range systemically happened in both two chronologies, it means the uncertain age affect limited on their relative age. And that is very important for our study. The OSL paper (Li et al., 2018) discovered the climate variation between Central Asia and East Asia presented an anti-phase pattern on precession component during the last interglacial. It means the climate change in Central Asia is about 10 ka lagging to East Asia on precession component. Although the length of lagging time is different, considered the uncertainties range of chronology, the result of OSL chronology is not conflicted with our result. According to comments, we will add some published OSL dating data in central Asia to discuss the unparallel variations between these two regions during Last interglacial. Nevertheless, thanks for your comments.

Reference: Chen, F. H., Jia, J., Chen, J.H., Li, G.Q., Zhang, X.J., Xie, H.C., Xia, D.S., Huang, W., and An, C.B.: A persistent Holocene wetting trend in arid central Asia, with wettest conditions in the late Holocene, revealed by multi-proxy analyses of loesspaleosol sequences in Xinjiang, China, Quat. Sci. Rev., 146, 134-146, 2016. Ding, Z.L., Ranov, V., Yang, S.L., Finaev, A., Han, J.M., and Wang, G.A.: The loess record in southern Tajikistan and correlation with Chinese loess, Earth Planet. Sci. Lett., 200, 387-400, 2002. Frechen, M., and Dodonov, A.E.: Loess chronology of the Middle and Upper Pleistocene in Tajikistan, Geol. Rundsch., 87, 2-20, 1998. Guo, Z.T., Berger, A., Yin, Q.Z., and Qin, L.: Strong asymmetry of hemispheric climates during MIS-13 inferred from correlating China loess and Antarctica ice records, Clim. Past, 5, 21-31, https://doi.org/10.5194/cp-5-21-2009, 2009. Hao, Q.Z., Wang, L., Oldfield, F., Peng, S.Z., Qin, L., Song, Y., Xu, B., Qiao, Y.S., Bloemendal, J., and Guo, Z.T.: Delayed build-up of Arctic ice sheets during 400,000-year minima in insolation variability, Nature, 490, 393-396, 2012. Li, G.Q., Chen, F.H., Xia, D.S., Yang, H., Zhang, X.J., Madsen, D., Oldknow, C., Wei, H.T., Rao, Z.G., Qiang, M.R.: A Tianshan Mountains loess-paleosol sequence indicates anti-phase climatic variations in arid central Asia and in East Asia. Earth and Planetary Science Letters, 494, 153-163, 2018. Stevens, T., Buylaert, J.P., Thiel, C., Újvári, G., Yi, S., Murray, A.S., Frechen. M., Lu, H.: Ice-volume-forced erosion of the Chinese Loess Plateau global Quaternary stratotype site. Nature Communications, 9, 983, 2018. Sun, Y.B., Clemens, S.C., An, Z.S., Yu, Z.W.: Astronomical timescale and palaeoclimatic implication of stacked 3.6-Myr monsoon records from the Chinese Loess Plateau. Quaternary Science Reviews, 25, 33-48, 2006. Zhang, J.J., Li, S.H., Sun, J.M., Hao, Q.Z.: Fake age hiatus in a loess section revealed by OSL dating of calcrete nodules. Journal of Asian Earth Sciences, 155, 139-145, 2018.
* * *

---

## Author Comment (AC1) · 3 Feb 2020

The study by Jia et al. deals with an interesting loess-palaeosoil section in Central Asia (Darai Kalon) with the concern to reconstruct palaeoenvironmental conditions for the last four glacial-interglacial cycles. Furthermore, the results are compared to a loess section from the Chinese Loess Plateau (Xifeng) in order to find spatial and temporal relations between humidity periods and atmospheric circulation patterns, especially related to the interaction between westerlies, the Indian Summer Monsoon and the East Asian Summer Monsoon. In its present form, this study reveals some weak points that mainly relate to the methodological approach for establishing chronologies for the

loess sections. (I) First, no independent age determination is presented that could support the chronology that is solely based on the assumed relation between grain sizes and ïĄď18O values from marine cores. Thus, the whole chronology remains very speculative. If the chronologies of loess sections from different areas are both established based on a peak matching between the grain-size curves and one ïĄď18O-curve, of course the resulting curves are synchronous - that is circular reasoning.

Response: As our response to Dr. Ujvari's comments, we agree it is better to construct the chronology with a high-resolution, absolute independent dating method in paleoclimate reconstruct studies. However, if we hope the age error of mid-Pleistocene loess is less than one thousand years, or even less, it is not easy. OSL dating method is a good choice to dating the Holocene aeolian deposit, especially for loess. If the material is good enough, the OSL age can limit the error in 200 years. However, for Last Interglacial loess, only 1% error means a 2 thousand years (ka) uncertainties range! For MIS11 loess, only 1% error means a 8 ka uncertainties range! I must say your work (Stevens et al., 2018) is an excellent work on OSL dating. However, even it, the age of Last Interglacial loess also still has an about 5 ka uncertainties range, or even more! As our conclusion, the lagging activity of moisture variation in Central Asia is mostly less than 5 ka. Therefore, the quality of OSL dating is not good enough for our story. That is why we had not selected OSL dating in this study. As reviewer mentioned that "If the chronologies of loess sections from different areas are both established based on a peak matching between the grain-size curves and one ïĄď18O-curve, of course the resulting curves are synchronous", it is very important for our study. Due to our aim is to compare the paleo-precipitation variation patterns between westerlies dominated Asia and monsoon dominated Asia, we select grain-size proxy to establish the relative chronology, which is independent with paleo-precipitation (Sun et al., 2004) and has close link with global ice volume (Hao, et al., 2015). And the synchronous variations on orbital scales between Tajikistan loess and Chinese loess had been proposed and demonstrated by paleomagnetic data (Ding et al., 2002).

References: Ding, Z.L., Ranov, V., Yang, S.L., et al., 2002. The loess record in southern Tajikistan and correlation with Chinese loess. Hao, Q.Z., Wang, L., Oldfield, F., 2015. Extra-long interglacial in Northern Hemisphere during MISs 15-13 arising from limited extent of Arctic ice sheets in glacial MIS 14. Scientific Reports, 5, 12103. Stevens, T., Buylaert, J.P., Thiel, C., et al., 2018. Ice-volume-forced erosion of the Chinese Loess Plateau global Quaternary stratotype site. Nature Communications, 9, 983. Sun, D.H., Bloemendal, J., Rea, D.K., et al., 2004. Bimodal grain-size distribution of Chinese loess, and its palaeoclimatic implications. Catena, 55, 325-340.

(II) Second, the direct correlation between grain-size values / magnetic parameters and the Specmap-isotope curve implies that the built-up of the curves (in the sense of the built-up of the loess-palaeosoil sequences from bottom to top) is a continuous, successive and more or less linear process. But loess sedimentation rates are highly variable and may even tend towards zero during interglacial periods (see e.g. the study by Frechen and Dodonov (1998) at the same section, cited in the manuscript). That means that phases of lacking (or strongly reduced) loess deposition that relate to periods of pedogenesis (that may extend over several thousands of years) are ignored in the age model in favor of producing a kind of continuous "loess graph" assuming uniform deposition rates between the time control points. In my opinion, beside all other uncertainties of the chronology, already this linear interpolation between a few age control points is a strong contradiction and produces large errors. This is all the more serious if the interpretation strongly relies on assumed time lags in the range of a few thousand years. Furthermore, pedogenic clay formation that leads to finer grain sizes in subsoil horizons is a secondary process that is decoupled from primary aeolian deposition, but it strongly affects the mean grain-size curve. The approach to directly correlate the gradual fining of the grain-size in a loess depth profile with a linear progression of time (e.g. sedimentary ïĄd'18O-curves, SPECMAP) is far away from the general concept of the way in which soils are formed. In my opinion, this also applies to magnetic susceptibility curves as soon as it is assumed that increasing values are caused by pedogenic processes instead of sedimentary processes. Such an approach

could be just about acceptable for the aim to e.g. make a general charcterisation of soil formation or for realizing a rough comparison or correlation of different soils from different sections, but not for generating a precise chronology for pedogenically altered loess sections.

Response: Firstly, the study by Frechen and Dodonov (1998) exhibited an age gap during 120-96 ka. It must be caused by systemically younger dating and the unignored error of estimated dose rates (Zhang et al., 2018) evidenced by the thick carbonate illuvium. And the stratigraphy do not support such hiatus (Ding et al., 2002; Bronger, 2003). Actually, the recent OSL dating study suggested the interglacial loess sediment presented a stable sediment rate in central Asia (Li et al., 2018). Secondly, if the leaching of fine-grained material in strata can obviously influence the mean grain size of the sublayer, both two loess records should present the unparallel variations and the lagging changed magnetic parameter. However, our data exhibited XF section (Chinese loess), developed the strongly pedogenic soil units, records the synchronous variations between grain size and magnetic parameters, and DK section (Tajikistan loess), developed the weakly pedogenic soil units, records a lagging change of the magnetic parameter. It strongly supports the lagging changed magnetic parameter in the DK record is not caused by the secondary process as the reviewer mentioned.

References: Bronger, A., 2003. Correlation of loess-paleosol sequences in East and Central Asia with SE Central Europe: towards a continental Quaternary pedostratigraphy and paleoclimatic history. Quaternary International, 106-107, 11-31. Ding, Z.L., Ranov, V., Yang, S.L., et al., 2002. The loess record in southern Tajikistan and correlation with Chinese loess. Li, G.Q., Chen, F.H., Xia, D.S., 2018. A Tianshan Mountains loess-paleosol sequence indicates anti-phase climatic variations in arid central Asia and in East Asia. Earth and Planetary Science Letters, 494, 153-163. Zhang, J.J., Li, S.H., Sun, J.M., Hao, Q.Z., 2018. Fake age hiatus in a loess section revealed by OSL dating of calcrete nodules. Journal of Asian Earth Sciences, 155, 139-145.

(III) Third, as I understand it, the interpretation of time lags is strongly based on the respective position and the possible offset between peaks of the grain-size and frequency dependent susceptibility curves. Or alternatively, if a palaeosoil shows a thicker subsoil horizon and a less rapid clay decline with depths, also the frequency-dependent susceptibility may show a less rapid decline as these values are generally strongly correlated with clay contents. This may lead to a different or shifted slope of the susceptibility curve. In that case the assumed time lags are again caused by pedogenic processes, i.e. different soil development depths. Therefore, despite value curves/ peaks of different sedimentological/pedogenetic parameters appear in different soil depths, both may relate to the same formation time. Without showing the proxy-curves in a clear stratigraphic context, it is not possible to evaluate the plausibility of the used approach. Furthermore, it would be necessary to show the strength of correlation between clay and frequency-dependant susceptibility (FDS) values, because if they show a strong correlation, FDS is not an independent value and thus even more severely influenced by the tuning process.

Response: Actually, we do not use the offset between peaks of the grain-size and frequency-dependent susceptibility curves to determine the lagging variations of paleo-precipitation in Tajikistan, but to compare the offset between peaks of frequency-dependent susceptibility curves between Tajikistan and Chinese loess records. The grain size parameter is only used to construct the chronology. So, we have not faced the problem the reviewer mentioned.
* * *

---

## Author Comment (AC2) · 3 Feb 2020

In this manuscript, the authors compare the magnetic susceptibility records between the DK and the Xifeng sections and conclude that the moisture variation in central Asia lagged the one in monsoon Asia by 0-5 kyr during four interglacials. Understanding the phase relationship between climates of different regions is extremely important for improving our understanding of the forcing and feedback mechanisms, and this kind of study should be encouraged. However, the analysis of phase relationship between two records has a high demand on the quality and accuracy of the chronology of the records. In this study, there are weaknesses in chronology that prevent me to be convinced of the results. Firstly and most critically, age uncertainty is not mentioned in the paper. As far as I know, the age uncertainty in loess records can reach several kyr. How would the conclusion of the authors be influenced by the age uncertainty of the two loess records? Secondly, the chronology of both DK and Xifeng sections was developed by correlation with the marine oxygen isotope record. This is quite acceptable in many loess studies, but I feel it is questionable when discussing phase relationship because the chronology of the two records is not independent. Lastly but less concerned, the mechanisms given by the authors to explain the phase difference between the two records is unclear for me. Response: As the reviewer mentioned, the absolute independent dating usually presents thousands or more than ten thousands error, that is why we selected the relative dating method to construct chronologies of two sections. And it is important, some absolute independent dating result had published in previous studies which provided a base to construct the chronology. We believe the grain size parameter is mostly independent of frequency-dependent susceptibility. If they are not independent, they will vary in-phase or anti-phase. However, our data exhibited DK loess record (Tajikistan loess) presents a lagging change of magnetic parameter, which is different from the XF section (Chinese loess) record. In the Chinese Loess Plateau, all loess records exhibit a synchronous variation between mean grain size and frequency-dependent susceptibility on the orbital scale, and that is independent with the pedogenic intensity of soil units (Hao et al., 2012). For the last question, due to the reviewer have not pointed out where is the problem, I cannot reply.

References: Hao, Q.Z., Wang, L., Oldfield, F., et al., 2012. Delayed build-up of Arctic ice sheets during 400,000-year minima in insolation variability, Nature, 490, 393-396.